# The *Jaws* Brachioplasty: An Original Technique: Improving Aesthetic Outcomes in Arm Lift Procedures

**DOI:** 10.3390/jcm11175038

**Published:** 2022-08-27

**Authors:** Giuseppe Nisi, Francesco Ruben Giardino, Martino Giudice, Giorgio Fasano, Roberto Cuomo, Luca Grimaldi

**Affiliations:** Plastic and Reconstructive Surgery Unit, Department of Medicine, Surgery and Neuroscience, University of Siena, 53100 Siena, Italy

**Keywords:** brachioplasty, upper limbs, post-bariatric surgery, body contouring

## Abstract

(1) Background: The increase in the number of bariatric surgery procedures has led plastic surgeons to look for new approaches to improve outcomes of body-contouring surgeries. A major concern in brachioplasty is the scarring process. Here, we propose a novel technique to minimize the incidence of pathological or unsatisfactory scars from brachioplasty. A video of the entire procedure is provided. (2) Methods: From January 2016 to August 2020, we performed the “Jaws” brachioplasty on 16 post-bariatric patients. We evaluated the effectiveness of the technique through pre- and postoperative assessments by patients and surgeons, the Vancouver Scar Scale, and the detection of major and minor complications within 12 months of follow-up. (3) Results: Thirteen patients were female and three were male, with a mean age of 32.5 ± 6.8 years (range: 22–47 years). The BODY-Q© Arms Section scores improved significantly, with no incidence of major or minor complications over 1 year of follow-up, and favorable aesthetic outcomes. (4) Conclusions: We believe that the “Jaws” technique is a valid contribution to post-bariatric surgery, as it aims to solve specific aesthetic problems of scarring from brachioplasty. The small number of patients does not allow the comparison of our original technique to others previously described in the literature.

## 1. Introduction

Brachioplasty is a surgery developed to correct aesthetic deformities of the arms in patients who have large segments of loose tissue (“bat wings”) on the upper limbs. Obese patients who have had significant weight loss are becoming increasingly important in numbers, and these patients are major candidates to correct aesthetic deformities related to loose skin. Significant and rapid weight loss results in numerous problems, such as excess skin, the deformity of various parts of the body, and tissue ptosis, which very frequently cause a high degree of discomfort for the patient. In this context, body contouring procedures, including brachioplasty, find their maximum expression with the aim of restoring harmony to the body and alleviating the patient’s psychological discomfort.

The American Society for Aesthetic Plastic Surgery estimates that a total of 24,662 brachioplasty procedures were performed in 2016, about four times the number of procedures performed in 2001 [1]. Moreover, the American Society for Metabolic and Bariatric Surgery estimates that about 200,000 bariatric surgery procedures were performed in 2016, predicting a further increase in these interventions in the coming years [2]. Based on these data, it is easy to assume that the demand for the body contouring procedure will increase exponentially in the coming years.

Brachioplasty is a surgical procedure that requires careful preoperative planning considering the characteristics and extent of the deformity of the arms in order to achieve the best results for the patient and minimize the risk of complications. Different classifications are used in the pre-intervention evaluation phase to select the most suitable treatment for the individual patient. Among these, the most frequently used are those proposed by Teimourian and Malekzadeh [3], Appelt et al. [4], and Elkhatib [5].

Brachial remodeling procedures are not new. In the 1930s, Thorek popularized an elliptical excision of the medial part of the arm [6]. Correa-Iturraspe and Fernandez in 1954 focused on the aesthetic refinements of the intervention [7]. In the 1970s, the addition of a Z-shaped plastic was tested to solve scar contraction problems, and in the same decade, Pitanguy added an extended approach of resection of the excess skin of the arm and also of the armpit through a single incision [8]. In 1995, Lockwood published an article regarding the superficial fascial system as a support structure of the arm and its role in brachioplasty, concepts that have become key principles of this procedure [9].

Traditional brachioplasty with scar placement over the bicipital groove, with or without extension to the lateral chest wall, is one of the most commonly used approaches. An alternative to the placement of the scar on the bicipital groove is brachioplasty with a posterior or postero-medial scar [10,11,12,13,14]. The incision patterns can be different, resulting in a straight-line scar, a T-shaped scar, a W-shaped scar, an L-shaped scar, or an S-shaped scar. These can be associated with a Z-shaped axillary plastic to prevent skin retraction [8,15,16,17,18,19,20]. In order to reduce the tension in the suture line, to improve the quality and visibility of the scar, and to reduce its migration, El Khatib proposed a brachioplasty with a posteromedial incision to which suspension points to the deep fascia are added [21]. The large number of practicable techniques, the heterogeneity of postoperative results, and the surgeon’s preferences contribute to the difficulty in choosing which technique is superior to the others.

Despite the continuous technical evolutions and the attention paid to improving the results, brachioplasty is a surgical procedure that frequently results in complications. These include wound dehiscence, seroma, lymphocele/lymphedema, difficulty in arm adduction, pathological scarring, infection, bleeding, nerve compression, compartment syndrome, neuromas, and sensory loss [22]. Dehiscence of the wound and hypertrophic scarring are the most frequent complications due to the incision of the brachioplasty, which is perpendicular in relation to the Langer lines, and the particularly thin skin of the arm. All of this seems amplified in obese patients, who have poor tissue quality.

All the brachioplasty techniques described since the first one by Thorek focus on the quality of the scar. In our opinion and experience, scarring is one of the major issues of body contour surgery. To improve scar quality and reduce tension on the suture line, we experimented with a new technique: the “Jaws” brachioplasty. Starting from Lockwood’s idea, we thought to use dermal flaps to shift the tension to a deeper and tougher plane (the dermis). We also wondered if the suspension of these flaps away from the suture line could help in reducing the incidence of hypertrophic scarring and dehiscence of the wound. The resulting shape of dermal flaps is where the “Jaws” brachioplasty derives its name.

## 2. Materials and Methods

### 2.1. Patients

A retrospective analysis was conducted on all patients undergoing brachioplasty with the “Jaws” technique at the Department of Plastic Surgery of Santa Maria alle Scotte University Hospital, Siena, Italy, between January 2016 and August 2020. The inclusion and exclusion criteria for patients undergoing surgery with this new technique are summarized in Table 1.

Epidemiological data of the patients, with particular attention to age, BMI, and previous surgical history, were collected.

The effectiveness of the technique was assessed through different methods. The first was through the assignment of a score by three different surgeons (one involved in the operation as the first operator and two surgeons who did not participate in the operation) based on pre- and postoperative photos. This score was elaborated on 4 evaluation items (arm shape and harmony with respect to the forearm; scar quality; correction of the initial deformity; overall evaluation) using a score scale from 1 to 6, where 1 represents a poor result and 6 is the most desirable result. It was also evaluated through the administration, before and after surgery, of a questionnaire (BODY-Q© Arms Section [23]) to the patients, through the assessment of scarring with the Vancouver Scar Scale (V.S.S.), and finally through the identification of major and minor complications after a 12-month follow-up. The BODY-Q© Arm Section is a scientifically sound patient-reported outcome instrument that is used worldwide to measure outcomes in patients who undergo weight loss and/or body contouring (and upper arm lift, in particular). It gives a measure of a patient’s own perception of the global aesthetic of the upper arm (e.g., size, shape, tone of the arm). It is made up of 7 items, and each one is scored from 1 to 4 points. The Vancouver Scar Scale is an objective evaluation made by the physician. It considers the vascularity, pigmentation, pliability, and height of scars. We also took note of possible complications occurring during the one-year follow-up (wound dehiscence, seroma, lymphocele/lymphedema, difficulty in arm adduction, pathological scarring, infection, bleeding, nerve compression, compartment syndrome, neuromas, and sensory loss). Statistical analysis was performed using Student’s t-test, with the aid of IBM^®^ SPSS^®^ Statistics 25.

### 2.2. Preoperative Drawing

Preoperative drawing was conducted on patients in an upright position with the upper limbs abducted by 90 degrees. A line connecting the apex of the axillary cord and the medial condyle of the humerus was drawn. By upper and lower “pinch and gather” maneuvers, the extent of the resection area was roughly estimated and demarcated at the skin level by enclosing it within two curves lines, one upper and one lower than the line drawn previously. These two lines, which unite to form a lozenge, represent the upper and lower incision lines. Three or four equilateral triangles with the side centered on the lower incision line were delimitated within the predicted skin excision area.

The triangles end up being adjacent to each other along the whole lower incision line (see Figure 1). These triangles are de-epithelialized, elevated, and fixed during the surgical procedure.

### 2.3. Surgical Technique

Surgery was performed under general anesthesia. The patient was placed in the supine position with the arms abducted at 90°.

The first operative phase consists of the infiltration, by means of a cannula, of the Klein solution through two micro-incisions placed at the ends of the cutaneous lozenge drawn in the preoperatory stage and in the subsequent lipoaspiration conducted exclusively in the area delimited by the skin to be removed. A re-evaluation of pre-surgical drawings is then carried out, which is eventually adjusted to the amount of residual skin following emptying due to lipoaspiration.

The perimeter of the skin to be removed is incised and dissected from the subcutaneous plane, safeguarding the adipose tissue that is not incised or affected by surgical removal. Skin is the only layer which must be excised. The respect of underlying adipose tissue and lymphatic network is a paramount target in this procedure, which must be executed carefully. This reduces complications such as lymphorrea, seroma, and lymphoedema of the upper arm. Care is taken to leave three/four triangular dermal flaps, as per the preoperative drawing, with the apex facing the anterior edge of the arm and having the lower edge of the incision as a base. These dermal flaps are lifted by dissection from the subcutaneous plane (Figure 2).

At this point, a fundamental phase of the surgery is represented by the reconstruction, corresponding to the projection of the bicipital groove, of the continuity of the subcutaneous adipose tissue and the synthesis of its bicipital and lower tricipital portion. The suture of the two adipose compartments, separated from each other by the down-migration due to the sudden and important weight loss, takes place with reversed stitches with a 3/0 absorbable braided thread and allows us to reconstruct the natural convexity of the volar surface of the arm, with an optimization of the aesthetic result.

After the reconstruction of the subcutaneous adipose tissue, a subdermal dissection of three/four triangular areas specular to the dermal flaps previously prepared is carried out, starting from the upper incision line. The dermal flaps are subsequently pulled under the triangular areas obtained and fixed through transfixed skin stitches in non-absorbable material (Figure 3).

Lastly, a two-plane suture is performed: subcutaneous reversed stitches with 3/0 Vicryl^®^ positioned exclusively at the ends of the triangular dermal flaps, and an intradermal suture with 4/0 Monocryl^®^ sealed by Dermabond^®^. A moderate compression dressing is applied. A photo was taken for each patient 12 months after surgery (see Figure 4).

A video showing pre-operative markings and surgical procedure is provided (see Appendix A).

## 3. Results

From January 2016 to August 2020, sixteen patients underwent brachioplasty with the “Jaws” technique. The patients included thirteen women and three men. The mean age was 32.5 ± 6.8 years (range: 22–47 years) and the mean BMI was 27.25 ± 1.69 Kg/m² (range: 24.6–29.8 Kg/m²). All patients enrolled in the study were post-bariatric surgery patients: six underwent sleeve gastrectomy, eight underwent gastric bypass, and two underwent OAGB. The patients’ mean weight loss was 49.5 ± 15.3 Kg (range: 22–73 Kg). Patients’ characteristics are summarized in Table 2.

Examining the results of the evaluation, three independent observers defined the results as good, with every surgeon giving each of the four items examined an average score greater than 4. The item that earned the lowest score from all three surgeons was the “scar quality”, which received an average score of 4. Despite the low average score, if we consider the individual patients, again with regard to the quality of the scar, average results higher than 3.5 were always obtained, with the exception of two patients with an average valuation of 3.

Evaluating for each patient the sum of the individual items of the BODY-Q© Arms Section, shown in Table 3, we note a statistically significant increase in the values between the preoperative and postoperative scores (t-value = 11.69 *p* < 0.0001), with an average increase of 11.3 points.

No statistically significant relationship was found between patients’ weight loss and pre or postoperative BODY-Q© score.

Two patients (Pt. 4 and Pt. 8) with the lowest judgment regarding the item “scar quality” recorded two of the greatest increases in scores between pre and postoperative BODY-Q© (+15 and +14, respectively).

The Vancouver Scar Scale demonstrated typical scores below 6/13, considered the limit of sufficiency, except for two patients who presented with aesthetically unsatisfactory scars. Except for unsatisfactory scarring, no other minor or major complications occurred during the one-year follow-up.

## 4. Discussion

The substantial impact of obesity on public health has led to the development and spread of bariatric surgery and complementary disciplines.

Post-bariatric surgical techniques fall under the domain of plastic surgery, with a unique and increasingly wide population of patients (as evidenced by the large number of techniques and tricks tailored on brachioplasty alone). Typical deformities of the arms (“bat wings”) causing aesthetic distress in obese patients must be corrected through an upper arm lift, giving rise to an increased demand for this type of operation.

The major concern for the surgeon is the scarring process due to intrinsic limitations of the surgical technique. The brachioplasty incision, which allows the excision of an adequate amount of skin excess in obese patients, is perpendicular to the Langer line. Therefore, the scar is continuously stretched. Our approach, with the “Jaws” brachioplasty, aims to reduce the tension on the surgical wound through transfixed points. This shifts tension forces from the incision line to the insertion point of the dermal flaps. We applied to brachioplasty one of the basic principles of plastic surgery: reducing tension on the scar. In addition, the two triangular dermal flaps of the proximal and distal extremities allow, in our opinion, a partial lifting effect on the skin of the elbow area and an axillary portion. Using a classic approach, these two areas are often hard to lift properly.

This new dermal flap concept in brachioplasty allows us to reduce the tension from the inside of the upper arm. Reducing tension on the scar is what is commonly achieved with steri-strips, compressive medications, and sheaths, but these are temporary and removable devices, which, however, we suggest to use in addition to “Jaws” brachioplasty.

The preparation of the dermal flaps makes it possible to reduce the number of subcutaneous stitches. This allows us to minimize traumatism and foreign bodies underneath the scar. The result is a lower incidence of diastasis of the surgical wound, a typical complication of brachioplasty. The three or four transfixed points at the base of the technique, despite the expectations, did not generate visible decubitus on the skin six months after surgery.

The success of this operation is tied to the execution of a precise preoperative design. Moreover, we believe that the use of continuous intradermal resorbable sutures, accurate hemostasis, adequate liposuction, excision of the skin layer while respecting the underlying adipose tissue and lymphatic network, follow-up, and elastic and compressive medications may help achieve favorable aesthetic results, as evidenced by the significant increase in the BODY-Q© score. This significant improvement (*p* = 0.0001) alone proves the efficacy of the technique in terms of aesthetic outcome, as perception of the body is a key element in obese patients to be taken into consideration. Moreover, the scores resulting from an independent evaluation of the outcomes by three specialists can be considered good, as an average overall score of 4.75/6 can be observed.

Less brilliant scores resulted from the evaluation of the scar, although with an average score higher than 3.5/6 in 14 cases out of 16. The Vancouver Scar Scale score was satisfying in 14 cases out of 16 (V.S.S. > 6 points). This means that scar issues in body contour surgery, specifically in obese patients, remain. The poor quality of the skin in massive weight loss patients and the typical incision lines of brachioplasty (perpendicular to the skin tension lines) are currently non-changeable factors.

Given the favorable outcomes we obtained in all the patients examined, we believe that this technique has wide applicability, most likely in patients who did not undergo bariatric surgery. The use of liposuction and superficial dissection of the skin flap to be removed make the technique safe. In fact, our cases did not reveal nerve damage (no sensitive or functional sequelae one year after surgery).

With the present technique, we achieved a null rate of complications in our small series, significant improvement in upper arm aesthetic perception by the patients, and acceptable scar outcomes.

We believe that the subjectivity of the BODY-Q© score is not a limit, but an asset to our research, as it globally analyzes the impact of the surgical correction of upper arm deformities on a patient’s life. It appears that there is no correlation between the patient’s perception and the surgeon’s perception. This may be due to severe improvements in BODY-Q© score in all patients and due to the different items evaluated by patients and surgeons. Remarkably, patients 4 and 8, with the lowest judgement of the scar by the surgeons, recorded the most increased BODY-Q© scores (+15 and +14, respectively).

In our preliminary report of 16 patients, our results indicate an excellent profile of efficacy and safety. However, it is difficult to establish the superiority of this technique compared to conventional techniques. The major problem which makes comparison nearly impossible, due to ethical and purely procedural reasons, is the inability to perform two different approaches on the same patient. Even the objective evaluation of scars is difficult to standardize between different study groups, different techniques, and examiners. Given this, different methods for assessing scars have been proposed since the 1990s. The use of pneumatometers, derma-spectrometers, tissue ultrasound palpation systems (TUPS), and Laser Doppler perfusion imaging are some of the tools designed specifically for the objective measurement of scar assessment items. As some studies have shown, each method has problems of reproducibility or practicality. Arguably, in the near future, technology will provide the tools for a standardized and objective evaluation of scarring outcomes, so as to conclusively define the effectiveness of the various techniques [24,25,26,27].

## 5. Conclusions

The data collected in our study are strongly suggestive of the efficacy and safety of the “Jaws” brachioplasty. This technique allows the tension to shift away from the scar through the dermal flaps described. The proximal dermal flap (axillary side) and the distal one (elbow side) can also help in properly lifting the skin in the axillary and elbow regions. These solutions have never been described before among brachioplasty refinements and techniques. The high satisfaction of patients and the overall evaluation by three independent surgeons suggest the achievement of suitable aesthetic outcomes using this technique. Some concerns with the scarring process, however, remain due to the poor quality of the skin in massive weight loss patients and the orientation of the brachioplasty incision. The small number of patients does not allow the comparison of the data presented in this study with more ample clinical records and statistics, especially concerning the incidence of major and minor complications and pathological or unsatisfactory healing. However, we believe that the presented technique is a valid contribution to post-bariatric surgery, as it aims to solve specific aesthetic problems of brachioplasty.

We finally emphasize the need to combine the “Jaws” brachioplasty with other post-bariatric surgery techniques and specific expertise due to the specific peculiarities of patients [28] and the difficulties of their healing paths.

## Figures and Tables

**Figure 1 jcm-11-05038-f001:**
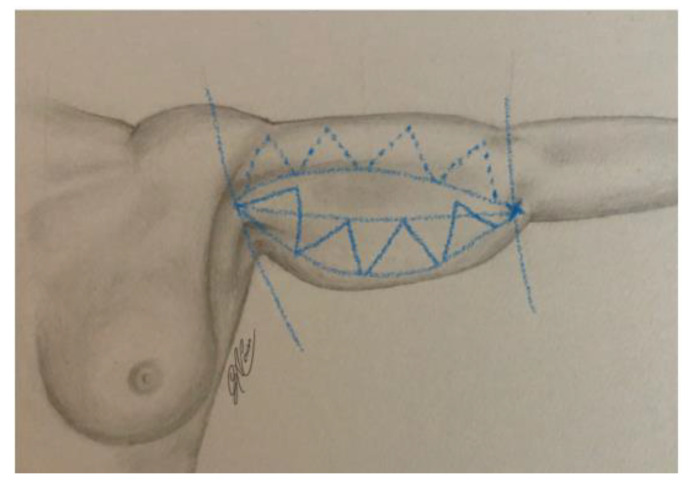
Preoperative drawing.

**Figure 2 jcm-11-05038-f002:**
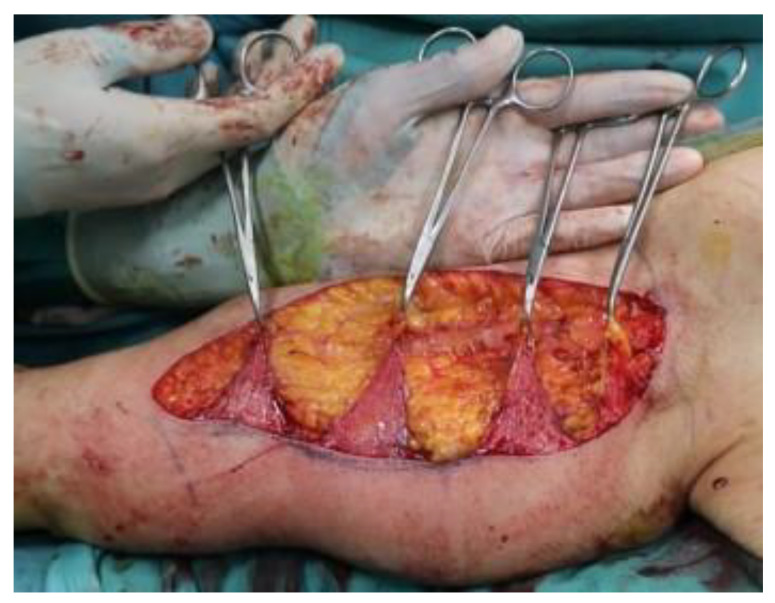
The dermal flaps. The shark jaw-like appearance of this step of our surgical procedure led us to name the technique the “Jaws brachioplasty”.

**Figure 3 jcm-11-05038-f003:**
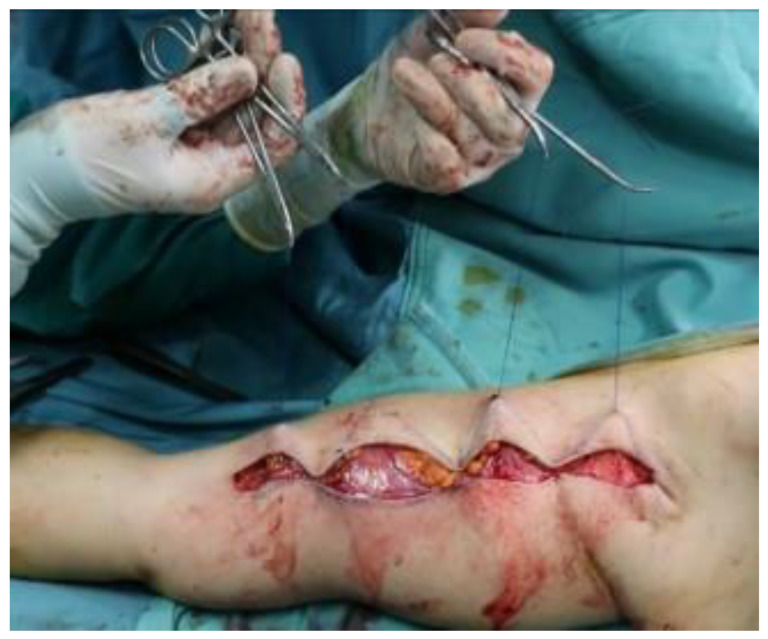
Fixation of dermal flaps with transfixed skin stitches.

**Figure 4 jcm-11-05038-f004:**
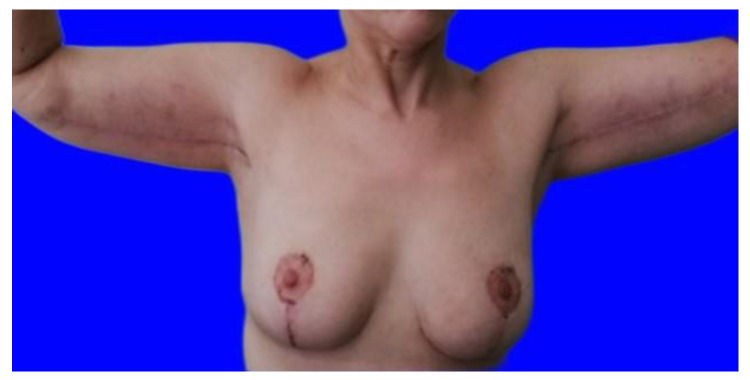
A patient 12 months after the intervention.

**Table 1 jcm-11-05038-t001:** Inclusion and exclusion criteria.

Inclusion Criteria	Exclusion Criteria
Obese patients who have undergone at least one bariatric surgery (OAGB, gastric bypass, sleeve gastrectomy, gastric banding, biliopancreatic diversion, gastric balloon) with stable weight for at least 12 months Pittsburgh Score > 10 BMI < 30	Patients with diabetes Patients who smoke more than 15 cigarettes per day Patients with unstable weight in the last 12 months Patients who have previously undergone brachioplasty Patients with coagulopathies, known scarring diseases, vasculitis

**Table 2 jcm-11-05038-t002:** Patients’ characteristics.

	SEX	AGE	BMI	Surgical Procedure	Weight Loss
**Patient 1**	F	29	25	Sleeve gastrectomy	67
**Patient 2**	F	29	27.3	Sleeve gastrectomy	34
**Patient 3**	F	34	29.4	Gastric bypass	52
**Patient 4**	F	22	29.8	Gastric bypass	50
**Patient 5**	F	43	25	Gastric bypass	70
**Patient 6**	F	47	27	Gastric bypass	28
**Patient 7**	F	32	24.6	One anastomosis gastric bypass	73
**Patient 8**	F	30	24.9	Gastric bypass	65
**Patient 9**	F	30	27.5	Sleeve gastrectomy	55
**Patient 10**	F	33	28	Sleeve gastrectomy	55
**Patient 11**	F	23	28.3	Gastric bypass	42
**Patient 12**	F	37	28.9	Sleeve gastrectomy	46
**Patient 13**	F	39	26.4	One anastomosis gastric bypass	31
**Patient 14**	M	25	26.1	Gastric bypass	63
**Patient 15**	M	28	28.8	Sleeve gastrectomy	40
**Patient 16**	M	39	29	Gastric bypass	22

**Table 3 jcm-11-05038-t003:** Difference between the sum of the preoperative and postoperative BODY-Q© items.

	Preoperative BODY-Q© Score	Postoperative BODY-Q© Score	Difference between Preoperative and Postoperative Score
**Patient 1**	8	18	+10
**Patient 2**	11	21	+10
**Patient 3**	11	25	+14
**Patient 4**	7	22	+15
**Patient 5**	13	24	+11
**Patient 6**	8	23	+15
**Patient 7**	12	23	+11
**Patient 8**	11	25	+14
**Patient 9**	9	21	+12
**Patient 10**	15	26	+11
**Patient 11**	13	19	+6
**Patient 12**	8	21	+13
**Patient 13**	7	15	+8
**Patient 14**	8	18	+10
**Patient 15**	9	21	+12
**Patient 16**	13	22	+9
**Mean Score**	10.1	21.5	+11.3 (*p* <0.0001)

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
