# Peer review of "The Jaws Brachioplasty: An Original Technique: Improving Aesthetic Outcomes in Arm Lift Procedures"

_jcm, 2022, doi:10.3390/jcm11175038_

Round 1

Reviewer 1 Report (Previous Reviewer 1)

The present version of the manuscript is much improved from the earlier versions in all the sections. The concerns raised have been considered. My only concern is that there is no etihical statement from the authors regarding the study. Was there an approval taken before carrying out this study. If yes then this needs to be included.

Author Response

Dear reviewer,

thank you for your valuable suggestion. We added, at the end of the article, a brief explanation regarding the ethical committee statement. 

We look forward your eventual further suggestion to improve the quality of our article.

Kind regards

Reviewer 2 Report (New Reviewer)

The authors describe a variation of the incision pattern of Brachioplasty, which they call the "Jaws Brachioplasty". While the technique has some novelty to it, I believe the manuscript has certain flaws that need to be addressed. 

1. There is no control group or reference group in this study. I concur with the authors, that it would not be feasible to operate on one patient with two different techniques. However, it would at least have been possible to gather retrospective data on scar quality and patient satisfaction from patients that have received Brachioplasty before the Jaws modification was introduced. It is therefore not possible, to for example state that the technique reduces the incidence of diastasis of the surgical wound because no comparison with standard technique has actually been made. Without a controll group, the data does not fulfill the definition of a retrospective study and cannot be described as such in the text. 

2. The number of patients is relatively low with 16 patients included. I think it would be more appropriate to describe this as a case series instead of a retrospective study (see also point 1). 

3. The authors firmly establish correlation without considering confounding factors. For example, they state that the present technique achieved a null rate of complications, but the absence of complications could be caused by numerous reasons such as the low number of cases or the comparatively rigorous exclusion criteria (no diabetic patients and no smokers). Statements like that have to be rephrased. 

3. Overall the paper is written with many repetitions and would highly benefit from shortening. For instance, in the introduction historic techniques and scar patterns are described twice. 

4. There are too many figures and tables. In particular table 3 and 4 do not add significant value to the paper but puff up the material (see also point 3). 

5. The authors state that the transfixed points of the dermal flaps did not generate visible decubitus on the skin. They can however clearly be seen in figure 4 and in the supplementary video. The authors may choose to describe the additional scarring as not significant to the patients, but stating that it is not there is clearly a missleading statement. 

6. Inserting the Jaws film poster into the supplementary video to explain the name the authors gave their Brachioplasty variation is not objective and deflects attention from the scientific value of the proposed technique. It would also be nice, if the text inserted into the video was english instead of italian. 

Author Response

Dear Reviewer,

we proceeded to fix most of the concerns you kindly expressed to improve the quality of our article. We shortened the lenght of the introduction, avoiding repetitions, deleted the redundant Table 3 and fixed some minor linguistic issue. In the manuscript we re-submit to you, we avoid definitive sentences about the "no complication technique" (point 3 of your review). 

We could easily fix the video, re posting a new-one. 

We look forward your valuable suggestions.

Kind regards

Round 2

Reviewer 2 Report (New Reviewer)

The authors have improved their manuscript. While it still might benefit from text editing by a native speaker, I do believe that it is of interest to readers. 

I hope the authors include a corrected version of the video in the final version with English titles, as mentioned in their response.

This manuscript is a resubmission of an earlier submission. The following is a list of the peer review reports and author responses from that submission.

Round 1

Reviewer 1 Report

In the present manuscript Nisi et al have proposed a novel technique for brachioplasty. In general the concept and idea of the technique is very interesting. The paper fails to make an impact on the reader because of the complicated style of writing and unclear sentences. A detailed review is mentioned below.

Major concerns

The title of the paper states ‘ Improving aesthetic and functional outcomes in arm lift procedures’ although no  functional outcomes were mentioned in the manuscript.

In the introduction the authors need to add a paragraph on the different techniques being followed in the post bariatric surgery patients and why there is a need to introduce a new method. In the abstract (line 21) the authors mention ‘ as much as it aims to solve specific aesthetic and functional problems of brachioplasty’. A brief description of these ‘aesthetic and functional problems’ caused by the other techniques can be mentioned along with how the present technique resolved these issues.

The discussion section needs extensive revisions as it is very hard to follow.

The manuscript will benefit from english editing, as many of the sentences are long and too complicated to understand. The initial sentence of the discussion is itself very long and the meaning of the sentence is lost in between. It is always preferred to have smaller sentences with a clear impact.

The sentence “Post Bariatric surgery is a discipline of the Plastic Surgery’ is not clear, it could be better written as – ‘post bariatric surgical techniques fall in the domain of plastic surgery ‘.

Another example is the sentence “Moreover, the preparation of the dermal flaps has made possible to reduce the number of subcutaneous ( subcutaneous what?), minimizing traumatism on the scar and ensuring greater tightness on the scar and reduced incidence of diastasis of the surgical wound, a typical complication of brachioplasty”, which is long and incomplete.

As this is a novel technique it would be interesting to know the background of the concept of their technique, using dermal flaps. Has this technique or a similar one has been used previously? The authors need to mention the advantages and disadvantages of the different brachioplasty techniques in the post bariatric surgery patients and how their method has an advantage.

The authors used questionnaires in their study. The results of the different questionnaires need to be discussed clearly. It is not sufficient to say that they were good or bad, rather the authors need to clarify the reason why this was ‘good or bad’. A brief decription of the questionnaires and their utility, example BODY-Q, Vancouver Scar Scale, could help the reader understand what they were used to assess.

What was the main reason behind getting lower scores for the scars by the surgeons? The authors need to mention the different issues with the scarring. This will help in future applicability of the technique.The post operative complications, even though minimal, also need to be mentioned.

Minor concerns

What does the Pz stand for – it could be better to mention it once as a merged row and then the patient numbers underneath. Mention the total score vertically and horizontally in Tables 3 and 4 with the p value, wherever applicable.

In line 166 mention the Table number.

The words “former obese” used in the discussion are better sounding than the use of the terms “ex-obese” and can be interchanged throughout the manuscript

Reviewer 2 Report

Thank you for the opportunity to review the manuscript. It describes the experimental technique - the Jaws brachioplasty. I do not see the approval of Institutional Review Board, which in my opinion is obligatory for such work. The article is interesting, it presents the novelty, the attached mp4 file is well-prepared, nevertheless I can see a few points that need to be improved.

  1. IRB
  2. I do not like the statement 'ex-obese', nor the literature does. Obesity is a disease similar to alcoholism. Patient with normal weight, who was obese, is still the patient with the obesity.
  3. The authors do not report prebariatric weight and BMI. In my opinion it is mandatory to report weight loss and %EWL or %TWL.
  4. Is there any correlation between weight loss/%EWL and BodyQScale/Vancouver Scar Scale? Have you measured the patients satisfaction after brachioplasty and its correlation with weight loss as well?
  5. Did patients with unsatisfactory scarring have lower Body Q Scale score?

Round 2

Reviewer 1 Report

The authors have not added any value or carried out any changes to the manuscript that were requested before. There was no introduction of the different brachioplasty procedures in post bariatric surgery patientsas suggested. 

The Tables are not in the standard format and neither were the total of the columns or P values provided. 

In their response the authors have mentioned that "it is impossible to say advantages and disadvantages of different brachioplasty", but then they have prove the novelness of their technique compared to previous ones. 

The manuscript in the present format is not suitable for publication and can only be considered after the authors, seriously make extensive changes to all the sections.  

Author Response

Dear reviewer,

We kindly inform you that we performed extensive revisions on the manuscript, highlighted with “Track Changes” function. We changed extensively or partially every section of the manuscript.

In particular, we focused on changing table format, inserting the P value in the columns, along with the amount of weight-loss. In the introduction, we focused on different types of brachioplasty (especially on W scar and Z scar brachioplasty) and, above all, the concept of fascial system by Lockwood. We quoted and highlited in the text the best evidences already present in literature regarding brachioplasty procedures and complications, We modified extensively the discussion and the conclusion. We revised the English language, changed or removed the term “ex-obese”. We tried to discuss better why and how we came to Jaws brachioplasty, explained better the innovations introduced and why our technique may be considered among others. 

Please let me know if further revisions are required.

Reviewer 2 Report

I cannot agree with the authors that reporting weight loss after bariatric surgery, even if it is plastic surgery paper, is not needed. Moreover, as the authors noticed, it contributes to brachioplasty effect. 

I understand what the used scales are. And I still think that the authors can measure the correlation between the opinions of patients and physicians. 

I don't feel the authors improved the manuscipt after the review or did not understand the review. 

Author Response

Dear reviewer,

We kindly inform you that we performed extensive revisions on the manuscript, highlighted with “Track Changes” function. In particular, we focused on changing table format, inserting the P value in the columns, along with the amount of weight-loss. In the introduction, we focused on different types of brachioplasty (especially on W scar and Z scar brachioplasty) and, above all, the concept of fascial system by Lockwood. We tried to discuss better why and how we came to Jaws brachioplasty, and explained better the innovations introduced. This led us to modify extensively the discussion and the conclusion. We discussed also  a possible correlation among patient and physicians. We revised the English language, changed or removed the term “ex-obese”.

Please inform us if further revisions are required.